Predictors of county-level diabetes-related mortality risks in Florida, USA: a retrospective ecological study

Deb Nath Nirmalendu
Odoi Agricola aodoi@utk.edu
Biomedical and Diagnostic Sciences, University of Tennessee , Knoxville , TN , United States of America
Gould Gwyn
Electronic publication date: 2025 Jan 16
Publication date: 2025
Volume: 13
Electronic Location ID: e18537
Received 2024 Jul 2; Accepted 2024 Oct 25
Copyright: ©2025 Deb Nath and Odoi
Copyright year: 2025
Copyright holder: Deb Nath and Odoi
License: This is an open access article distributed under the terms of the Creative Commons Attribution License, which permits unrestricted use, distribution, reproduction and adaptation in any medium and for any purpose provided that it is properly attributed. For attribution, the original author(s), title, publication source (PeerJ) and either DOI or URL of the article must be cited.
License URL: https://creativecommons.org/licenses/by/4.0/

Keywords: Diabetes, Predictor, Risk factor, Mortality risk, Mortality rate, Ecological study, Geographic Information Systems, GIS, Linear regression, Ordinary least squares regression

Funding: The authors received no funding for this work.

==============================
Background

Diabetes is an increasingly important public health problem due to its socioeconomic impact, high morbidity, and mortality. Although there is evidence of increasing diabetes-related deaths over the last ten years, little is known about the population level predictors of diabetes-related mortality risks (DRMR) in Florida. Identifying these predictors is important for guiding control programs geared at reducing the diabetes burden and improving population health. Therefore, the objective of this study was to identify geographic disparities and predictors of county-level DRMR in Florida.

Methods

The 2019 mortality data for the state of Florida were obtained from the Florida Department of Health. The 10th International Classification of Disease codes E10-E14 were used to identify diabetes-related deaths which were then aggregated to the county-level. County-level DRMR were computed and presented as number of deaths per 100,000 persons. Geographic distribution of DRMR were displayed in choropleth maps and ordinary least squares (OLS) regression model was used to identify county-level predictors of DRMR.

Results

There was a total 6,078 diabetes-related deaths in Florida during the study time period. County-level DRMR ranged from 9.6 to 75.6 per 100,000 persons. High mortality risks were observed in the northern, central, and southcentral parts of the state. Relatively higher mortality risks were identified in rural counties compared to their urban counterparts. Significantly high county-level DRMR were observed in counties with high percentages of the population that were: 65 year and older (p < 0.001), current smokers (p = 0.032), and insufficiently physically active (p = 0.036). Additionally, percentage of households without vehicles (p = 0.022) and percentage of population with diabetes (p < 0.001) were significant predictors of DRMR.

Conclusion

Geographic disparities of DRMR exist in Florida, with high risks being observed in northern, central, and southcentral counties of the state. The study identified county-level predictors of these identified DRMR disparities in Florida. The findings are useful in guiding health professionals to better target intervention efforts.

Introduction

Diabetes affects millions of people worldwide. The number of diabetes patients in the United States (US) has been increasing over the last two decades, and it is projected to double or triple by 2050 (Boyle et al., 2001). As of 2021, 38.4 million people in the US had the disease, of whom 29.7 million were diagnosed, while 8.7 million were undiagnosed and unaware of their illness (Centers for Disease Control and Prevention , 2022). The condition is closely associated with other chronic diseases such as heart disease, kidney disease, hypertension, stroke, and cardiovascular disease (Adailton da Silva et al., 2018; Centers for Disease Control and Prevention, 2023c; Centers for Disease Control & Prevention, 2023e; Khan, Uddin & Srinivasan, 2018; Zinman et al., 2017). Therefore, the risk of death is higher among people with diabetes compared to those without the condition (Centers for Disease Control and Prevention, 2023c; Centers for Disease Control and Prevention, 2023b). Diabetes is the eighth leading cause of death in the US (Centers for Disease Control and Prevention, 2023d), and a total of 103,294 people died from the disease in 2021 (Centers for Disease Control & Prevention, 2023e). However, recent studies indicate that the impact of diabetes on overall mortality is significantly underestimated (Stokes & Preston, 2017; Xu et al., 2022). In reality, diabetes mortality risk in the US is nearly 12%, implying that it is the third leading cause of death in the nation, after heart disease and malignant neoplasms (Stokes & Preston, 2017; Xu et al., 2022).

Fifteen Southeastern states, including Florida, have higher prevalence of type 2 diabetes (≥11.0%) than the nation’s average (8.5%) (Danaei et al., 2009; Baker et al., 2011). In 2011, the Centers for Disease Control and Prevention (CDC) declared the 644 counties of those 15 states as “diabetes belt” (Baker et al., 2011). In Florida, the prevalence of the disease is estimated at 12.5% and has been increasing over the past ten years (Florida Department of Health, 2017). The economic cost of the disease in Florida was estimated at approximately $25 billion ($19.3 billion direct costs and $5.5 billion indirect costs) in 2017 (Florida Department of Health, 2017). It is reported that medical expenses for individuals with diabetes in Florida are 2.3 times higher than for those without diabetes (Florida Department of Health, 2017).

Despite advances in diabetes management and treatment, the mortality risk associated with diabetes remains high in Florida. It is estimated that the age-adjusted diabetes-related mortality risk (DRMR) in the state has increased from 16.9 to 24.2 per 100,000 persons over the last ten years (Florida Department of Health, 2023a). Although some previous studies used rigorous epidemiological/statistical approaches to investigate pre-diabetes and diabetes prevalence (Khan et al., 2021; Lord, Roberson & Odoi, 2020; Lord, Roberson & Odoi, 2023; Okwechime, Roberson & Odoi, 2015), very little is known about DRMR and associated predictors in Florida. Therefore, the objective of this study was to identify county-level predictors of DRMR in Florida. This information will be useful for guiding control efforts and would contribute towards achieving one of the objectives of the Healthy People 2030 of reducing health disparities and enhancing overall population health.

Methods

Ethics approval

This study was approved by the University of Tennessee Institutional Review Board (IRB number: UTK IRB-23-07809-XM). The review board determined that the study “is eligible for exempt review under 45 CFR 46.101 pursuant to category 4ii: Secondary research for which consent is not required: Secondary research uses of identifiable private information or identifiable biospecimens, if information, which may include information about biospecimens, is recorded by the investigator in such a manner that the identity of the human subjects cannot readily be ascertained directly or through identifiers linked to the subjects, the investigator does not contact the subjects, and the investigator will not re-identify subjects”. The authors did not have access to information that could identify participants during or after data collection.

Study area

This study was conducted in the state of Florida which is located between 27°66′N and 81°52′W and spans 65,758 square miles. County land area ranges from 243.6 square miles (Union County) to 1,998 square miles (Collier County) (United States Census Bureau, 2023a). As of 2022, Florida was one of the most populous states in the US. About 22.2 million people live in the state, 50.8% of whom are female and 49.2% are male (United States Census Bureau, 2023b). Miami-Dade County is the most populous (2.6 million people), whereas Liberty County, located in the northern part of the state (Fig. 1), is the least populated with only 7,987 people (United States Census Bureau, 2023b). The age distribution of the adult population in Florida is 24% 18–34 years old, 26% 35–49 years old, 25% 50–64 years old, and 22% are seniors (≥65 years old) (United States Census Bureau, 2023b). There are 76.8% White, 17% Black, and 6.2% all other races in Florida. Of the 67 counties in the state, 30 are rural and 37 are urban (Fig. 1).

Figure 1 Study area showing the geographic distribution of rural and urban counties in Florida, USA.

Diabetes-related death data

Individual-level death data covering the time period January 1 to December 31, 2019, were obtained from the Florida Department of Health. The International Classification of Disease (ICD) 10th revision was used to identify the cause of death, and ICD-10 codes E10-E14 were used to identify diabetes-related deaths (World Health Organization, 2019). No distinction was made between type 1 and 2 diabetes. The county of residence of the deceased was used to aggregate the number of diabetes-related deaths to county-level using R statistical software version 4.2.2 (R Core Team, 2023). Population estimates for 2019 were obtained from the American Community Survey (ACS) (United States Census Bureau, 2022) and used as denominator to calculate DRMR. County-level DRMRs were then calculated and expressed as number of deaths per 100,000 persons. A conceptual model of the potential predictors of DRMR was constructed (Fig. 2). Data on potential predictors of DRMRs, for the year 2019, were extracted from several data sources (Table 1).

Figure 2 Conceptual model showing potential predictors of diabetes-related mortality risks.

Table 1 Data source and variables used in the identification of predictors of diabetes-related mortality risks in Florida, 2019.

Source	Data obtained	
Florida Behavioral Risk Factors Surveillance System (BRFSS)	Percentage of population with diabetes	
	Percentage of population attending DSMEa	
	Percentage of population reporting depressed	
	Percentage of population that are heavy drinkers	
	Percentage of population that have disability	
	Percentage of population with hypertension	
	Percentage of population with kidney disease	
	Percentage of population that get regular checkups	
	Percentage of population taking medication for cholesterol	
	Percentage of population with heart disease	
	Percentage of population that have had stroke	
	Percentage of population with arthritis	
	Percentage of population with insurance coverage	
	Percentage of population not going to a doctor for medical cost	
	Percentage of population that have personal doctor	
	Percentage of population reporting poor health	
	Percentage of population reporting good health	
	Percentage of population reporting normal weight	
	Percentage of population with obesity	
	Percentage of population who are overweight	
	Percentage of population that are underweight	
	Percentage of population that are current smoker	
	Percentage of population that are snuff users	
	Average age when diagnosed with diabetes	
	Percentage of population that are current e-cigarette users	
	Percentage of population with high cholesterol	
	Percentage of population taking medication for hypertension	
	Percentage of population eating vegetables once a day	
	Percentage of population eating fruits once a day	
	Percentage of population that are highly physically active	
	Percentage of population that are physically active	
	Percentage of population that are insufficiently physically active	
	Percentage of population that are physically inactive	
American Community Survey (ACS)	Percentage of population <20 years old	
	Percentage of population 20–44 years old	
	Percentage of population 45–64 years old	
	Percentage of population ≥65 years old	
	Percentage of population with income <25k per year	
	Percentage of population with income 25k–50k per year	
	Percentage of population with income >50k per year	
	Percentage of population that are non-Hispanic White	
	Percentage of population that are non-Hispanic Black	
	Percentage of population that are non-Hispanic Others	
	Percentage of population that are Hispanic	
	Percentage of population that have <high school education	
	Percentage of population with high school education	
	Percentage of population with some college education	
	Percentage of population with college education	
	Percentage of population that are married	
	Percentage of population that are divorced/widow/separated	
	Percentage of population that never married/unmarried couple	
	Percentage of population that are male	
	Percentage of population that are female	
	Percentage of households without vehicle	
Certified in Healthcare and Human Resources (CHHR)	Percentage of rural population	
	Average parts per million of CO2 or Air pollution	
	Percentage of population with limited access to healthy food	
	Percentage of population having food insecurity	
	Percentage of population not having access to exercise	
	Percentage of population that are unemployed	
United States Census Bureau TIGER Geodatabase	County-level cartographic boundary shapefile	
Notes.

a Diabetes Self Management Education.

All data are for 2019.

Descriptive statistics

Descriptive analyses were conducted using R statistical software version 4.2.2 (R Core Team, 2023) and implemented in R studio version 1.4.1717 (R Studio Team, 2020). Normal distribution of continuous variables was evaluated using the Shapiro–Wilk test. Since some of the continuous variables were not normally distributed, medians and interquartile ranges were used for summary statistics (Table 2).

Table 2 Summary statistics of potential predictors of county-level diabetes-related mortality risks in Florida, 2019.

Predictor	Mean	SD a	Median	Minimum	Maximum	IQR b	
Percentage of population with diabetes	13.37	3.09	12.9	6.4	20.8	2.3	
Percentage of population attending DSMEa	54.40	10.75	53.10	29.60	76.60	16.05	
Percentage of population reporting depressed	17.78	3.26	17.9	10.30	24.70	3.7	
Percentage of population that are heavy drinkers	7.22	2.24	7.00	1.30	12.20	2.85	
Percentage of population that have disability	34.46	5.38	34.60	21.00	45.90	7.85	
Percentage of population with hypertension	38.21	5.07	37.60	25.30	47.00	7.20	
Percentage of population with kidney disease	3.76	1.16	3.60	1.70	7.70	1.50	
Percentage of population that get regular checkups	76.12	3.88	76.10	63.20	89.10	4.85	
Percentage of population taking medication for cholesterol	61.33	5.12	61.40	47.60	70.50	7.60	
Percentage of population with heart disease	5.65	1.48	5.70	2.50	9.00	2.00	
Percentage of population that have had stroke	4.52	1.29	4.50	1.20	7.00	2.00	
Percentage of population with arthritis	28.97	5.31	28.70	17.80	40.20	6.90	
Percentage of population with insurance coverage	82.61	4.31	83.20	68.60	90.50	4.60	
Percentage of population not going to a doctor for medical cost	16.41	3.01	16.00	9.50	21.90	4.45	
Percentage of population that have personal doctor	73.65	5.12	74.40	57.60	86.00	6.80	
Percentage of population reporting poor health	22.55	4.86	22.60	8.60	33.10	7.15	
Percentage of population reporting good health	77.45	4.86	77.40	66.90	91.40	7.15	
Percentage of population reporting normal weight	29.71	5.27	29.60	19.40	43.90	6.20	
Percentage of population with obesity	32.46	6.06	32.20	18.20	48.10	7.85	
Percentage of population who are overweight	35.68	3.64	36.10	124.60	43.80	3.55	
Percentage of population that are underweight	2.14	0.91	2.10	0.30	5.40	1.25	
Percentage of population that are current smoker	19.14	5.08	18.50	11.00	32.40	6.65	
Percentage of population that are snuff users	4.98	3.27	3.60	1.20	13.50	5.10	
Average age when diagnosed with diabetes	48.94	2.49	49.10	42.40	53.50	3.05	
Percentage of population that are current e-cigarette users	5.73	1.84	5.70	2.00	13.10	2.15	
Percentage of population with high cholesterol	32.32	3.82	31.80	23.60	43.70	4.45	
Percentage of population taking medication for hypertension	78.95	4.07	78.83	67.32	89.25	4.95	
Percentage of population eating vegetables once a day	82.05	4.75	82.56	66.58	93.33	5.87	
Percentage of population eating fruits once a day	60.50	5.77	60.85	49.11	72.77	7.96	
Percentage of population that are highly physically active	34.53	5.61	33.78	24.35	54.60	7.00	
Percentage of population that are physically active	15.24	3.45	14.86	9.09	27.53	4.41	
Percentage of population that are insufficiently physically active	15.85	3.16	15.69	9.41	26.20	3.57	
Percentage of population that are physically inactive	34.35	0.28	33.58	22.67	51.23	10.68	
Percentage of population <20 years old	21.68	3.40	21.70	8.30	29.50	4.05	
Percentage of population 20–44 years old	29.98	5.19	30.80	13.90	41.50	6.10	
Percentage of population 45-64 years old	26.67	2.09	27.00	20.80	31.70	2.10	
Percentage of population ≥65 years old	21.64	7.73	20.10	11.60	56.70	8.25	
Percentage of population with income <25k per year	33.96	7.14	34.27	20.30	53.39	12.52	
Percentage of population with income 25k–50k per year	27.88	4.21	28.35	20.55	40.20	5.28	
Percentage of population with income >50k per year	38.15	9.41	37.17	19.39	58.94	16.29	
Percentage of population that are non-Hispanic White	69.75	15.02	74.07	13.07	89.53	15.68	
Percentage of population that are non-Hispanic Black	13.07	9.50	9.97	1.10	54.42	10.96	
Percentage of population that are non-Hispanic Others	3.83	1.65	3.64	0.92	8.59	2.05	
Percentage of population that are Hispanic	13.36	12.44	8.97	2.77	69.79	10.40	
Percentage of population that have <high school education	15.81	6.23	14.75	5.38	38.37	6.89	
Percentage of population with high school education	35.23	7.11	35.00	19.88	54.82	11.78	
Percentage of population with some college education	29.89	4.54	30.83	17.57	35.54	7.35	
Percentage of population with college education	19.07	8.06	18.44	6.06	35.62	13.31	
Percentage of population that are married	50.75	2.67	50.38	38.49	66.98	5.62	
Percentage of population that are divorced/widow/separated	24.19	3.28	24.85	16.34	30.61	4.46	
Percentage of population that never married/unmarried couple	25.06	5.79	23.88	10.60	45.16	7.26	
Percentage of population that are male	51.12	4.46	48.77	46.84	70.09	5.49	
Percentage of population that are female	48.88	4.46	51.23	29.91	53.16	5.48	
Percentage of rural population	37.50	32.26	23.77	0.02	100.00	58.63	
Percentage of households without vehicle	5.72	1.91	5.26	1.89	10.33	2.07	
Average parts per million or Air pollution	7.52	0.91	7.70	5.20	9.10	1.30	
Percentage of limited access to healthy food	9.33	5.74	9.00	0.00	31.00	6.00	
Percentage of food insecurity	14.00	2.22	14.00	10.00	20.00	3.50	
Percentage of not having access to exercise	68.94	24.52	77.00	10.00	100.00	35.00	
Notes.

a Standard Deviation.

b Interquartile Range.

Investigation of county-level predictors of DRMR

A global ordinary least squares regression (OLS) model was used to identify county-level predictors of DRMR in Florida. After selecting potential predictors using the conceptual model, a two-step process was used to fit a multivariable model with the outcome specified as DRMR. The first step in model-building was to assess the univariable associations between each potential predictor and the outcome. A liberal p-value of ≤0.20 was used for this assessment. Using p-value ≤0.20 allows for assessment of potentially important confounders during the multivariable analysis stage (Dohoo, Martin & Stryhn, 2012). All the variables that showed significant associations (based on a relaxed p-value of ≤0.20) in the univariable analyses were subjected to two-way Spearman rank correlation analyses. Spearman rank correlation analysis was appropriate for this assessment because some of the continuous variables were not normally distributed and Spearman correlation analysis does not assume normal distribution (Zar, 2014). When two variables showed a strong correlation (r > 0.7), only one of them was included in the subsequent multivariable model. The decision on which variable to include in the multivariable model from a pair of highly correlated variables was based on biological and statistical considerations. Backward elimination process was then performed, using a critical p-value of 0.05, to fit the final multivariable model. The backward elimination approach allows for assessment of confounders during the modeling process (Dohoo, Martin & Stryhn, 2012), enhances the accuracy of prediction, and reduces the likelihood of overfitting the data (Draper & Smith, 1998). Confounding variables were then evaluated by comparing changes in regression coefficients after running the model with and without the suspected confounder. If there was ≥20% change in the coefficients of any of the variables in the model, the suspected variable was then identified as a confounder and retained in the model regardless of its statistical significance. Two-way interaction terms of the variables of final multivariable model were evaluated based on biological relevance, and only those with a p-value ≤0.05 were included in the final model. The variance inflation factor (VIF) was used to assess multicollinearity in the final model. If the VIF value of a variable was ≥10, it was considered to have high collinearity with at least one of the other variables in the model. Overall goodness-of-fit of the final model was assessed using adjusted R-squared (R2) and Akaike Information Criterion (AIC). Residual plots, Jarque–Bera test, and Breusch-Pagan test were used to evaluate the assumptions of normality and homoscedasticity. The residuals were also used to identify outliers, while leverage, Cook’s distance, and Difference in Fits (DFITS) were used to identify influential observations. All the analyses were performed using R statistical software version 4.2.2 (R Core Team, 2023). Moran’s I, implemented in GeoDa (Anselin, Syabri & Kho, 2006), using 1st order queen contiguity spatial weights, was used to assess for spatial effects or spatial autocorrelation in the residuals.

Cartographic display

Cartographic boundary files were downloaded from the TIGER geodatabase (United States Census Bureau, 2023a) and used to generate maps. QGIS 3.34 (QGIS org, 2024) was used for all cartographic displays. Choropleth maps for the distribution of county-level diabetes-related mortality risks and significant predictors from the final multivariable model were generated using Jenk’s optimization classification scheme.

Results

Geographic disparities in distribution of diabetes-related mortality risks

There was a total of 6,078 diabetes-related deaths reported in Florida in 2019. Of these, 59.3% were male, 40.7% were female, 75.7% were White, 20.4% were Black, and 3.9% were all other races. The percentage of diabetes-related deaths was highest (69.6%) among seniors (≥65 years old). The county-level mortality risks varied across the state, ranging from 9.6 per 100,000 persons in St. Johns County to 75.6 in Desoto County (Fig. 3). Of the 67 counties in the state, 35 had mortality risks equal to or greater than the national average (31.1 per 100,000 persons). Counties with high mortality risks were mainly in the northern (Holmes, Washington, Jefferson, Taylor, Santa Rosa, and Dixie), central (Citrus, Hernando, Sumter, Marion, and Putnam), and south-central (Desoto and Hardee) parts of Florida (Figs. 1 and 3). Conversely, the southernmost counties (Broward, Collier, and Monroe) had relatively lower mortality risks compared to other parts of the state. It is worth noting that most counties with high mortality risks were mainly in rural areas.

Figure 3 Geographic disparities in distribution of diabetes-related mortality risks in Florida, 2019.

Predictors of diabetes-related mortality risks

Results of the univariable associations between county-level DRMR and its potential predictors are shown in Table 3. Forty-six of the 59 variables assessed had significant univariable associations with DRMR using a relaxed critical p-value of ≤0.20. However, based on the final multivariable model, significant predictors of disparities in county-level DRMRs were percentage of population that had diabetes (p < 0.001), were aged 65 or above (p < 0.001), were current smokers (p = 0.032), were insufficiently physically active (p = 0.036), and percentage of households without vehicles (p = 0.022) (Table 4). No significant interactions and confounding were detected. The p-value of Moran’s I (p = 0.747) indicated that there was no spatial effects or spatial autocorrelation in the residuals and hence the model did not violate the independence assumption. A comparison of the observed DRMR and the model estimates of the DRMR is shown in Fig. 4. The general pattern in distribution of the model estimates is quite similar to that of the observed DRMR indicating that the model is doing well in predicting DRMR in the study area (Fig. 4).

Table 3 Results of univariable associations between potential predictors and county-level diabetes-related mortality risks in Florida, 2019.

Predictors	Coefficient (95% CI a )	p-value	
Percentage of population with diabetes	−2.989 (2.069, 3.908)	<0.001	
Percentage of population attending DSMEb	−0.259 (−0.593, 0.073)	<0.124	
Percentage of population reporting depressed	−0.519 (−0.593, 1.631)	<0.354	
Percentage of population that are heavy drinkers	−1.595 (−3.173, −0.016)	<0.048	
Percentage of population that have disability	−1.328 (0.736, 1.921)	<0.001	
Percentage of population with hypertension	−1.329 (0.689, 1.969)	<0.001	
Percentage of population with kidney disease	−5.272 (2.417, 8.128)	<0.001	
Percentage of population that get regular checkups	−0.259 (−0.678, 1.196)	<0.583	
Percentage of population taking medication for cholesterol	−0.788 (0.102, 1.474)	<0.025	
Percentage of population with heart disease	−4.602 (2.420, 6.783)	<0.001	
Percentage of population that have had stroke	−4.926 (2.368, 7.483)	<0.001	
Percentage of population with arthritis	−1.587 (1.024, 2.149)	<0.001	
Percentage of population with insurance coverage	−0.624 (−1.456, 0.209)	<0.139	
Percentage of population not going to a doctor for medical cost	−1.627 (0.483, 2.772)	<0.006	
Percentage of population that have personal doctor	−0.239 (−0.469, 0.949)	<0.502	
Percentage of population reporting poor health	−1.351 (0.679, 2.023)	<0.001	
Percentage of population reporting good health	−1.351 (−2.023, −0.679)	<0.001	
Percentage of population reporting normal weight	−0.641 (−1.314, 0.033)	<0.062	
Percentage of population with obesity	−0.768 (0.197, 1.339)	<0.009	
Percentage of population who are overweight	−0.860 (−1.839, 0.119)	<0.084	
Percentage of population that are underweight	−1.073 (−2.921, 5.067)	<0.593	
Percentage of population that are current smoker	−0.674 (−0.024, 1.372)	<0.058	
Percentage of population that are snuff users	−1.110 (0.028, 2.192)	<0.045	
Average age when diagnosed with diabetes	−0.761 (−0.693, 2.214)	<0.300	
Percentage of population that are current e-cigarette users	−0.362 (−2.346, 1.622)	<0.717	
Percentage of population with high cholesterol	−1.559 (0.687, 2.431)	<0.001	
Percentage of population taking medication for hypertension	−0.798 (−0.086, 1.664)	<0.078	
Percentage of population eating vegetables once a day	−0.122 (−0.645, 0.889)	<0.751	
Percentage of population eating fruits once a day	−0.732 (−1.338, −0.126)	<0.019	
Percentage of population that are highly physically active	−0.118 (−0.767, 0.532)	<0.718	
Percentage of population that are physically active	−1.197 (−2.213, −0.181)	<0.022	
Percentage of population that are insufficiently physically active	−1.031 (−2.157, 0.096)	<0.072	
Percentage of population that are inactive	−0.641 (0.114, 1.168)	<0.018	
Percentage of population <20 years	−0.818 (−1.872, 0.236)	<0.126	
Percentage of population 20–44 years old	−0.804 (−1.477, −0.131)	<0.020	
Percentage of population 45–64 years old	−0.817 (−2.552, 0.917)	<0.350	
Percentage of population ≥65 years old	−0.575 (0.126, 1.025)	<0.013	
Percentage of population with income <25k per year	−0.695 (0.214, 1.176)	<0.005	
Percentage of population with income 25k–50k per year	−1.601 (0.831, 2.372)	<0.001	
Percentage of population with income >50k per year	−0.721 (−1.065, −0.377)	<0.001	
Percentage of population that are non-Hispanic White	−0.164 (−0.076, 0.403)	<0.177	
Percentage of population that are non-Hispanic Black	−0.081 (−0.303, 0.464)	<0.676	
Percentage of population that are non-Hispanic Others	−2.970 (−5.059, −0.882)	<0.006	
Percentage of population that are Hispanic	−0.234 (−0.521, 0.054)	<0.109	
Percentage of population that have <high school education	−0.850 (0.303, 1.396)	<0.003	
Percentage of population with high school education	−0.921 (0.461, 1.379)	<0.001	
Percentage of population with some college education	−1.059 (−1.818, −0.299)	<0.007	
Percentage of population with college education	−0.888 (−1.283, −0.493)	<0.001	
Percentage of population that are married	−0.009 (−0.684, 0.701)	<0.981	
Percentage of population that are divorced/widow/separated	−2.034 (1.042, 3.026)	<0.001	
Percentage of population that never married/unmarried couple	−0.659 (−1.268, −0.051)	<0.034	
Percentage of population that are male	−0.375 (−0.438, 1.188)	<0.360	
Percentage of population that are female	−0.375 (−1.187, 0.438)	<0.361	
Percentage of rural population	−0.131 (0.023, 0.239)	<0.018	
Percentage of households without vehicle	−1.654 (−0.211, 3.519)	<0.081	
Average parts per million of COb or Air pollution	−1.401 (−2.603, 5.404)	<0.487	
Percentage of population with limited access to healthy food	−0.664 (0.050, 1.278)	<0.034	
Percentage of population having food insecurity	−3.441 (2.039, 4.844)	<0.001	
Percentage of population of not having access to exercise	−0.209 (−0.348, −0.069)	<0.004	
Percentage of population of being unemployed	−11.581 (6.628, 16.535)	<0.001	
Notes.

a Confidence Interval.

b Diabetes Self Management Education.

Table 4 Results of multivariable model showing significant predictors of county-level diabetes-related mortality risks in Florida, 2019.

Predictors	Coefficient (95%CIa)	p-value	
Percentage of population ≥65 years old	0.009 (0.004, 0.014)	<0.001	
Percentage of population that are current smokers	0.008 (0.001, 0.015)	<0.032	
Percentage of population that are insufficiently physically active	0.013 (0.001, 0.026)	<0.036	
Percentage of households without vehicle	0.021 (0.003, 0.038)	<0.022	
Percentage of population with diabetes	0.034 (0.022, 0.046)	<0.001	
Notes.

a Confidence Interval.

Figure 4 Comparison of actual Diabetes-related Mortality Risk (DRMR) with model estimates of the DRMR.

The northern, central, and south-central counties had high percentages of population with diabetes (Fig. 5) and insufficiently physically active population, mirroring the spatial patterns of DRMRs (Fig. 3). High percentages of population with diabetes and insufficient physical activity tended to mainly occur in rural counties. The percentage of smokers tended to be high in the northern regions of the state, whereas high percentages of households without vehicles were evident across both northern and southern parts of the state (Fig. 5). Despite lower DRMR in the southernmost counties of the state, these regions had higher percentages of population aged 65 or above compared to other parts of the state (Fig. 5).

Figure 5 Geographic distribution of the predictors of diabetes-related mortality risks in Florida, 2019.

Discussion

This study investigated geographic disparities and predictors of county-level DRMR in Florida. The findings are crucial in identifying counties with high mortality risks and predictors of the identified spatial patterns so as to provide information for targeted evidence-based health programs to reduce DRMR in Florida.

The identified significant association between high county-level DRMR and high percentage of older population (age ≥65 or above) is consistent with findings of a previous study conducted by Dugani et al. (2022), who reported that county-level DRMR was significantly higher among the older population (greater than 55 years old) compared to younger adults. It is worth noting that Florida has the second-highest percentage of adults aged 65 or above (22%) (United States Census Bureau, 2023c). This demographic trend in Florida is associated with higher rates of diabetes-related comorbidities, resulting in higher DRMR in this older age group. Additionally, data from the Florida Department of Health (FDH) indicated that the mortality risk from diabetes among adults 65 or above (116.7 per 100,000 persons) was approximately 7.5 times higher than the mortality risk among those less than 65 years old (15.5 per 100,000 persons) (Florida Department of Health, 2023b).

The association between county-level percentage of physical inactivity and high DRMR observed in this study has been reported in several previous studies (Hu et al., 2004; Hamilton, Hamilton & Zderic, 2014; Tiang et al., 2023). One study reported that DRMR was 1.65 times higher among counties with high percentages physically inactive people compared to those counties with lower percentages of physically inactive people (Turi & Grigsby-Toussaint, 2017). This might be due to the fact that the counties with high percentages of physical inactivity are closely linked to higher risk of chronic conditions and shorter life expectancy (University of Wisconsin Population Health Institute, 2023). There is evidence that a large percentage of the population in counties with high DRMR do not meet the recommended physical activity guidelines compared to those in counties with low DRMR (Florida Department of Health, 2023c). This suggests the need to encourage the population to increase level of physical activity. Green spaces, including parks, trails, community gardens, and playgrounds, are important components of local built environment that impact physical activity and overall community health (Taylor et al., 2007). Therefore, increasing access to these green spaces and physical fitness areas would potentially help to increase the percentage of the population attaining the recommended level of physical activity (Centers for Disease Control and Prevention, 2023f) and potentially reduce the risk of chronic conditions including diabetes.

The observed positive association between the percentage of current smokers and DRMR is consistent with reports by the FDH that counties with percentage of current smokers higher than the national average (11.5%) experienced 1.3 times higher DRMR than those counties where the percentages of current smokers were below the national average (Florida Department of Health, 2023d). This might be due to the fact that most Florida counties with high percentage of current smokers are located in the northern rural regions of the state, where the educational attainment is generally lower, and the population might be unaware of the detrimental effects of smoking and diabetes (United States Census Bureau, 2023d). Smoking and diabetes are double hazards to the health of rural communities and aggravate diabetes complications, thereby increasing the risk of death (Zheng et al., 2020). Although not directly compared with the current study, several previous studies conducted at the individual level reported that smoking was an independent risk factor for diabetes-related mortality because of increased risk of other chronic conditions like cardiovascular disease and coronary heart disease among diabetes patients (Ford & De Stefano, 1991; Colhoun et al., 2001; Cederholm et al., 2008; Fagard & Nilsson, 2009; Campagna et al., 2019; Abdelhamid et al., 2023). It has also been reported that the risk of diabetes-related death is 1.6 times higher among smokers than non-smokers (Ford & De Stefano, 1991). Therefore, smoking cessation programs are important in reducing the risks of death from these conditions. The World Health Organization reported that smoking cessation not only reduces the risk of developing diabetes but also reduces the risk of diabetes complications and death (World Health Organization, 2023). Hence, continuing educational programs are encouraged for those counties with high percentage of current smokers and diabetes patients.

Reliable transportation plays a fundamental and crucial role in ensuring access to healthcare and medication. Persons with diabetes need reliable transportation to ensure regular clinician visits, access to medications, and adjustments in treatment plans as needed. The current study identified a significant association between high DRMR and high percentage of households without vehicles in Florida. Insufficient public transportation and long drive times to healthcare facilities in counties with a high percentage of households without vehicles might reduce accessibility to healthcare services (United States Department of Health and Human Services, 2022), leading to high DRMRs. A recent study conducted by Lord & Odoi (2024) at the zip code level in Florida also reported that lack of access to vehicles was significantly associated with diabetes-related hospitalization rates, thereby increasing the risk of complications associated with the disease. Furthermore, another previous study also reported that transportation barriers to healthcare facilities were more likely to be associated with South and Midwest counties compared to those from other parts of the country (Wolfe, McDonald & Holmes, 2020). As a result, understanding the relationship between county-level high DRMR and percentage of households without vehicles is important for addressing population health in most vulnerable regions of the state.

Strengths and limitations

This is the first study to investigate county-level predictors of diabetes-related mortality risks in Florida using rigorous statistical approaches. The findings of this study are useful for guiding evidence-based interventions by identifying DRMR disparities and its predictors. This study investigated county-level predictors of DRMR in Florida for one year (2019). It is possible that these associations change over time based on the changes in population socioeconomic and demographic profiles. Therefore, it is important that these investigations be part of regular health surveillance programs to provide the most current information to guide health planning and service provision. However, this study has some limitations. The BRFSS survey data are self-reported, and so might have reporting bias. The survey does not distinguish between type 1 and type 2 diabetes. Although type 1 and type 2 diabetes have different etiologies (type 2 diabetes are more influenced by lifestyle, socioeconomic, and behavioral factors, while type 1 diabetes is due to autoimmune disease), we only considered socioeconomic and demographic factors in this study. However, since the majority (90–95%) of diabetes in the United States is type 2, we expect the fact that there was minimal effect on the results of this study. Finally, since this is an ecological study conducted at the county-level, inferences must only be made at the county level and not individual level to avoid ecological fallacy. These limitations notwithstanding, the findings provide useful information regarding disparities and determinants of DRMR in Florida.

Conclusions

There is evidence that geographic disparities in DRMR exists and are determined by distribution of percentages of the population aged 65 or older, current smokers, population having insufficient physical activity, population with diabetes, and households without vehicles. These findings are important for guiding targeted health planning and service provision to reduce the disease burden and DRMR in Florida.

Supplemental Information

Supplemental Information 1 Raw data

The authors are grateful to the Florida Department of Health for providing the data.

Additional Information and Declarations

Competing Interests

Author Contributions

Human Ethics

Data Availability

Agricola Odoi is an Ecademic Editor for PeerJ.

Nirmalendu Deb Nath conceived and designed the experiments, performed the experiments, analyzed the data, prepared figures and/or tables, authored or reviewed drafts of the article, and approved the final draft.

Agricola Odoi conceived and designed the experiments, performed the experiments, authored or reviewed drafts of the article, and approved the final draft.

The following information was supplied relating to ethical approvals (i.e., approving body and any reference numbers):

University of Tennessee Institutional Review Board (IRB number: UTK IRB-23-07809-XM).

The following information was supplied regarding data availability:

The raw data are available in the Supplementary File.

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
