# Peer review of "Predictors of county-level diabetes-related mortality risks in Florida, USA: a retrospective ecological study"

_PeerJ, doi:10.7717/peerj.18537_

## Round 0.1 · original submission · Major Revisions

While both reviewers recognise the potential value of this study, there are some concerns that need to be carefully and fully addressed, notably from reviewer-1. Please address these fully and modify your paper accordingly.

Reviewer 1 ·

Basic reporting

In this study, Deb Nath and Odoi develop a predictive model of diabetes related mortality risk (DRMR) by performing an ecological, county level, statistical analysis on death rates and a large number of socioeconomic/health related predictors. The paper is well written, and the statistics are, for the most part, very well documented (my comment about the form of the model below aside). However, I believe that the study has methodological flaws in both the statistical modeling and the data sourcing that must be addressed. In particular, my two major comments are (1) the lack of consideration of spatial effects, which violates the independence assumption meaning that the model is fundamentally mis-specified, and (2) the combining of Type 1 and Type 2 diabetes, which have two fundamentally different etiologies.

Experimental design

Spatial Effects: The statistical methods are well reported and the resultant models were tested to confirm assumptions of normality and homoscedasticity. However, my major concern is that the assumption of independence has not been considered. In ecological models (indeed spatial models in general), there is, more often than not, a significant degree of spatial autocorrelation, violating the independence assumption – this is Tobler’s first law of Geography (Tobler, Econ Geogr, 1970, 10.2307/143141). In fact, the authors allude to the spatial correlation in the discussion when they group counties into “Northern”, “Central” etc. Spatial correlation can, and should, be tested for (e.g., using Moran’s I or the semi-parametric Mantel test). If spatial correlations exist and have not been accounted for in the model, then the model is mis-specified, affecting the results of statistical analyses and may result in spurious associations (see for example Shaikh et al., Ecol Indic, 2021, 10.1016/j.ecolind.2021.107992). Multiple ways exist to account for spatial autocorrelation (e.g., conditional autoregressive models such as the Besag-York-Mollié structure: Besag et al., Ann Inst Stat Math, 1991, 10.1007/BF00116466). The spatial element must be considered in the model structure.

Diabetes: I am concerned that Type 1 and Type 2 diabetes have been lumped together. From a medical perspective they are different diseases with different etiologies – there is no reason that the socioeconomic predictors of their mortality rate will be similar. Whilst both have environmental and genetic factors, broadly Type 1 is an autoimmune condition whilst Type 2 is predominantly influenced by lifestyle factors. The lifestyle factors that lead to Type 2 Diabetes (e.g., age, inactivity) are the same as the factors were determined to lead to increased DRMR, thus acting as massive confounders in the model when both are lumped together. The ICD-10 codes separate Type 1 (E10.X) and Type 2 (E11.X), so it is unclear why the authors have not done the same.

Additional Comments

Line 144: Please provide more detail on the original raw data. In particular how were the data aggregated at the county level – were location data at the individual level based on lived address or place of death (which could be the same but also in a hospital). This potentially has a large effect on the model (presence of large hospitals drawing patients across county lines) so it is important to discuss the potential effects. Proportion of patients with type 1 vs type 2 diabetes dying at home vs in care is also likely to be different (see my broader comment on combining the codes).
Table 1: Table 1 lists the variables sourced for the study. In the caption it is implied that these are all 2019 values but this is not clear, please clarify in the text that all the surveys are also from that year (e.g., ACS, BRFSS etc.).

Figure 2: It is great to see a conceptual DAG model in the paper – however, given its presence and complicated structure it is unclear how this is then translated into the form of the model. This should be clarified. In particular, are factors which are considered to only indirectly affect DRMR modeled correctly to avoid confounding colliders? I believe McElreath (2015, https://xcelab.net/rm/), in particular chapters 5 and 6, gives the clearest discussion of this in literature.

Validity of the findings

Limitations: The limitations of the study need to be expanded for full transparency. The given limitations and their implications should be discussed in more detail. There are further limitations that have not been discussed – some of these have been covered above, but namely the risk/loss of information provided by aggregating data (i.e., the risk of an ecological fallacy).

Minor comment: Fig 3 – it would be interesting to visually compare the modeled DRMR with the actual DRMR visually (e.g., in a panel plot).

·

Basic reporting

The paper is well written, in clear, professional English, and the text is easy to follow. The introduction and background do a good job of setting the context for the research question, and the objective is clearly identified. Some references could be added, or more details could be provided (see my annotations in the PDF). I suggest moving Table 2 and 3 and Figure 1 to the supplement.

Experimental design

The statistical analysis was appropriate, and it was reported adequately. Reasonable justifications were given for specific modeling choices. The raw data provided is sufficient for independent study reproduction.

Validity of the findings

The discussion is well-thought-out, concise, and focused on the results. Additionally, the results are biologically and theoretically plausible. Conclusions are made related to the study target population and are coherent with the study design.

Additional comments

Overall, the manuscript is thorough and well-written. I have two questions for the authors:
1. Why did the study span one year? Additionally, I would suggest including how that specific year might have impacted either the mortality or reporting of diabetes in the discussion.
2. Did you consider using multi-level modeling and/or a more granular representation of the sociodemographic variables used in the model?

---

## Round 0.2 · accepted · Accept

Thank you for taking the time to address the concerns of the reviewers. I now happy to recommend acceptance.

Reviewer 1 ·

Basic reporting

I am satisfied with the responses to my original review. Thank you for making the appropriate changes. I now believe that the manuscript is acceptable for publication.

Experimental design

I am satisfied with the responses to my original review. Thank you for making the appropriate changes. I now believe that the manuscript is acceptable for publication.

Validity of the findings

I am satisfied with the responses to my original review. Thank you for making the appropriate changes. I now believe that the manuscript is acceptable for publication.